# Series-Parallel Reconfigurable Electric Double-Layer Capacitor Module with Cell Equalization Capability, High Energy Utilization Ratio, and Good Modularity

Masatoshi Uno *, Ziyan Lin and Kakeru Koyama

College of Engineering, Ibaraki University, Hitachi 316-8511, Japan; 20nm665x@vc.ibaraki.ac.jp (Z.L.); 20nm620s@vc.ibaraki.ac.jp (K.K.)
* Correspondence: masatoshi.uno.ee@vc.ibaraki.ac.jp

**Abstract:** Voltages of electric double-layer capacitor (EDLC) modules vary rather wider than traditional secondary batteries. Although EDLCs should desirably be cycled in a voltage range as wide as possible to achieve a high energy utilization ratio, the wide voltage variation of EDLC modules impairs the performance of DC–DC converters. To address such issues, previous works reported series-parallel reconfiguration techniques, which are roughly divided into balance- and unbalance-shift circuits. However, conventional balance-shift circuits are not applicable to modules comprising odd number cells, impairing modularity. Unbalance-shift circuits, on the other hand, unavoidably cause cell voltage imbalance that reduces energy utilization ratio. This paper proposes a novel series-parallel reconfigurable EDLC module with cell voltage equalization capability. The proposed reconfigurable EDLC module is applicable to any number of cells, realizing good modularity. Furthermore, all cells in the proposed module can be charged and discharged uniformly without generating cell voltage imbalance, achieving an improved energy utilization ratio compared with conventional techniques. A five-cell module prototype was built for experimental verification. While the module voltage varied between 1.04 and 2.83 V, all cells discharged from 2.5 to 0.3 V. The result is equivalent to a 98.6% energy utilization ratio.

**Keywords:** electric double-layer capacitor (EDLC); energy utilization ratio; series-parallel reconfiguration; voltage equalization

## 1. Introduction

Energy storage devices, including lithium-ion batteries (LIBs) and electric double-layer capacitors (EDLCs), are indispensable in various applications, from small portable electronic devices to utility-scale systems. EDLCs offer various key advantages over traditional rechargeable batteries in terms of power density, cycle life, and operation temperature range. However, EDLCs cannot simply be an alternative to secondary batteries because of their low energy density properties. EDLCs have chiefly been used as a high-power-density energy buffer that supports the main battery [1,2]. Lithium-ion capacitors (LICs), a hybrid energy storage device of EDLCs and LIBs, have also been developed to enhance energy density [3–9]. Thanks to the long-life performance at a wide temperature range, EDLCs and LICs would expectedly replace traditional lead–acid batteries in infrastructure applications, where maintenance-free operation is strongly desirable. Another likely application is spacecraft energy storage systems, where traditional LIBs are cycled with shallow depth-of-discharge (DoD) to meet the requirement of 5 to 10 year operation. EDLCs and LICs can operate without significant degradation, even with deep DoD, hence dramatically bridging the energy density gap between LIBs and EDLCs/LICs [9,10].

Aside from the lower energy density properties, wide voltage variations of EDLCs and LICs are cited as a major drawback. Single-cell voltages of an EDLC, LIC, and LIB as a function of ampere-hour DoD are shown in Figure 1. The typical voltage variation ranges

of LIB cells are 2.7–4.2 V, whereas those of EDLCs and LICs are 0–2.5 V and 2.2–3.8 V, respectively. This suggests that DC–DC converters for EDCLs and LICs need to operate with a wide input/output voltage range. However, such wide input/output voltage operation generally impairs converters' performance, such as power conversion efficiencies and volume.

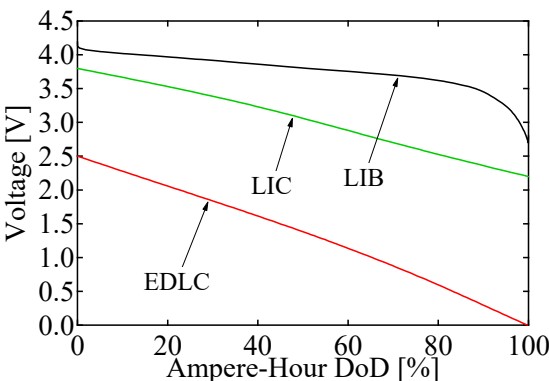

**Figure 1.** Cell voltage profiles of lithium-ion battery (LIB), lithium-ion capacitor (LIC), and electric double-layer capacitor (EDLC) cells as a function of ampere-hour depth-of-discharge (DoD).

EDLCs and LICs can be charged and discharged only in an allowable input/output voltage range of DC–DC converters. If an operational voltage range of a DC–DC converter is 1.0–3.0 V, an EDLC cell can be charged and discharged only in the range of 1.0–2.5 V. This implies that the energy stored in the voltage range of 0–1.0 V is no longer usable. In general, an EDLC's energy utilization ratio, $U$, is given by:

$$U = \frac{V_{cha}{}^2 - V_{cut}{}^2}{V_{cha}{}^2} = 1 - \left( \frac{V_{cut}}{V_{cha}} \right)^2 \tag{1}$$

where $V_{cha}$ is the charging voltage, and $V_{cut}$ is the cut-off voltage. A characteristic of $U$ as a function of $V_{cut}/V_{cha}$ is graphed in Figure 2. The lower the cut-off voltage $V_{cut}$, the higher the energy utilization ratio $U$ will be. For example, 75% of the stored energy can be utilized by discharging an EDLC cell as low as $V_{cut} = V_{cha}/2$. Discharging as low as $0.32 \, V_{cha}$ achieves 90% energy utilization. Wide input/output voltage range DC–DC converters achieve high energy utilization, but their performance is unavoidably impaired, as discussed above.

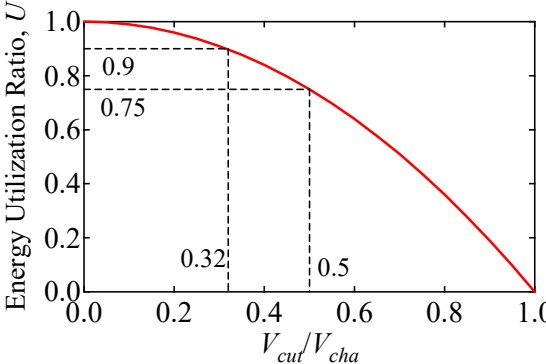

**Figure 2.** Energy utilization ratio $U$ as a function of $V_{cut}/V_{cha}$.

Series-parallel reconfiguration techniques, also known as changeover circuits, have been developed to cope with the above issues of EDLCs [11–18]. Literally, the series-parallel connections of EDLCs are reconfigured depending on cell voltages so that the module voltage stays within the desired voltage range. Figure 3 illustrates images of reconfiguration

sequence and voltage profiles of a four-cell module. In the discharging process (Figure 3b), the module starts to operate as a 1-series 4-parallel (1S-4P) configuration, with which the module voltage $V_M$ is equal to the cell voltage $V_C$. Before $V_M$ decreases down to the predetermined lower voltage limit $V_L$, the module is reconfigured to be 2S-2P configuration to double the module voltage $V_M$ (i.e., $V_M = 2\,V_C$). Before $V_M$ reaches $V_L$ again, the module is reconfigured to be 4S-1P configuration so that $V_M = 4\,V_C$. This reconfiguration technique allows cells to discharge deeply while keeping $V_M$ within a desired voltage range. In the charging process (Figure 3c), the series-parallel configuration is reconfigured in the opposite order.

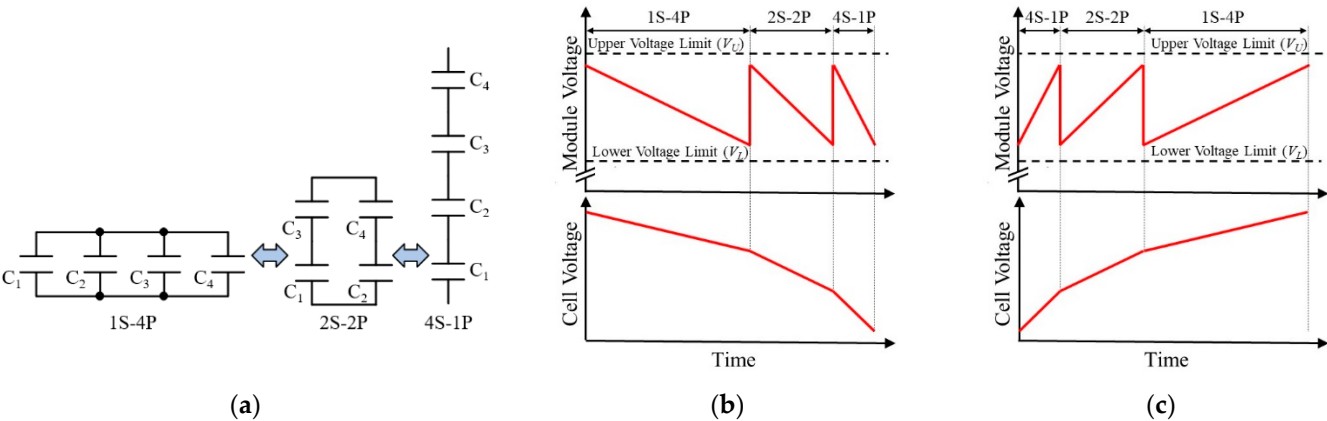

**Figure 3.** (**a**) Typical reconfiguration sequence of balance-shift circuit for four-cell module, and its (**b**) discharging and (**c**) charging profiles.

A variety of series-parallel reconfiguration modules have been proposed [11–18]. Depending on whether cell voltage imbalance occurs during series-parallel reconfiguration sequences, the conventional reconfiguration techniques can be categorized into two groups: balance- and unbalance-shift circuits. A typical balance-shift circuit consisting of four cells $C_1$–$C_4$ is shown in Figure 4a that operates identically to the circuit in Figure 3. This circuit operates either in the 1S-4P, 2S-2P, or 4S-1P configuration, and all cells can discharge and charge uniformly in all configurations. However, this technique cannot be used for modules comprising odd number cells (e.g., five and seven cells), and, therefore, its design flexibility and extendibility are poor. Employing cells with a larger/smaller capacitance is another way to tune the module capacitance, but the limited variety of products does not allow fine-tuning. For example, capacitances of DXE series from Nippon Chemi-Con corporation [19] are 400, 800, 1200, and 1400 F. Changing from 800 F cells to 400 F or 1200 F ones only achieves a 0.5 or 1.5 times greater module capacitance.

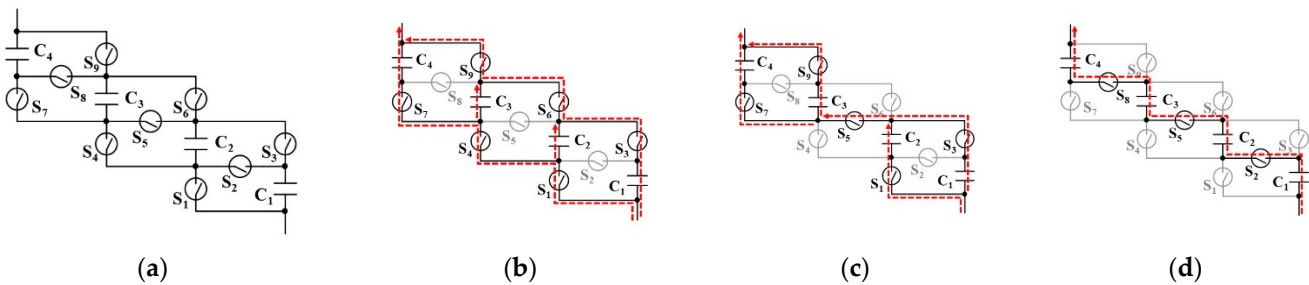

**Figure 4.** (**a**) Balance-shift circuit, (**b**) 1S-4P, (**c**) 2S-SP, and (**d**) 4S-1P configurations.

Representative reconfiguration steps and voltage profiles of the unbalanced-shift circuit for four cells $C_1$–$C_4$ are shown in Figure 5. $V_{C1}$–$V_{C4}$ in Figure 5b,c are the voltages of $C_1$–$C_4$. The number of series-connection changes one by one (i.e., 2S, 3S, and 4S), and voltage step changes can be finer than those of the balance-shift circuits. All cells can be

equally discharged and charged in 2S-2P and 4S-1P configurations, but unequal currents flow in the 3S configuration, generating cell voltage imbalance. In the discharging process, for example, $C_1$ and $C_4$ discharge faster, and $V_{C1}$ and $V_{C4}$ decrease steeper than $V_{C2}$ and $V_{C3}$. The occurrence of voltage imbalance significantly lowers the energy utilization ratio of the module, because some cell voltages decrease down to 0 V or some cells might be over-discharged under subzero voltages in the worst case. A representative unbalanced-shift circuit for four cells and its operation modes are shown in Figure 6 [17].

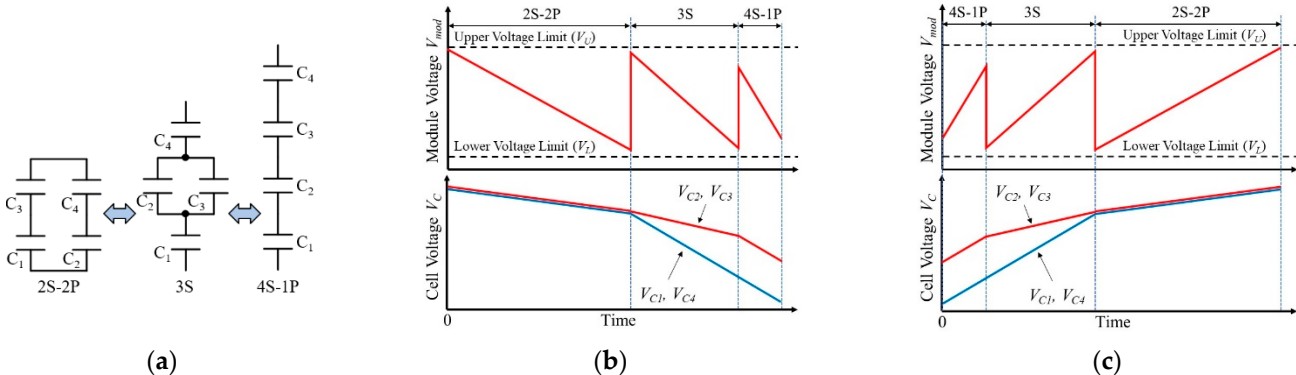

**Figure 5.** (**a**) Reconfiguration sequence of unbalance-shift circuit for four-cell module, and its (**b**) discharging and (**c**) charging profiles.

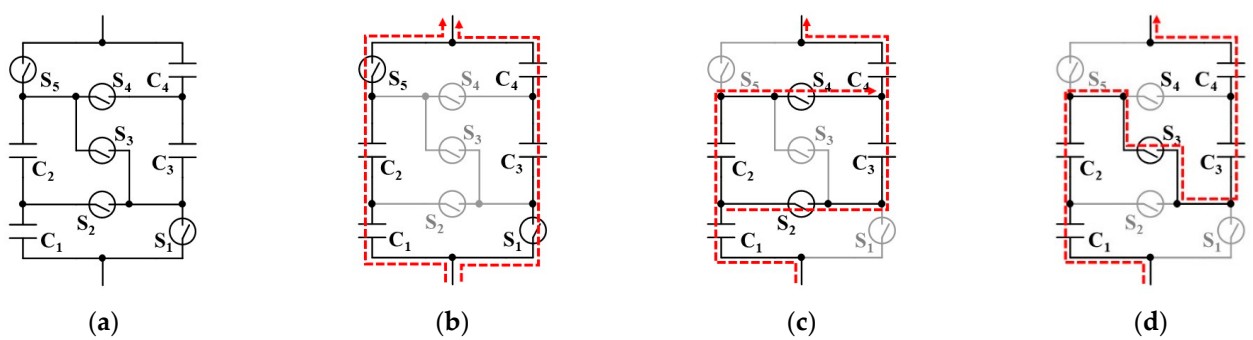

**Figure 6.** Unbalance-shift circuit for (**a**) four cells. Operation modes in (**b**) 2S-2P, (**c**) 3S, and (**d**) 4S-1P.

Previous work proposed the unbalance-shift circuit with cell voltage equalization capability [18]. All cells can charge and discharge uniformly, and the number of series-connection can be changed step-by-step. However, the minimum number of series-connection is limited to $n/2$ for $n$-cell modules—for example, the minimum number of series-connection of five- and six-cell modules is 3. The five-cell module can operate as either 3S, 4S, or 5S configurations, but 1S and 2S configurations are not feasible. This limited configuration results in a reduced energy utilization ratio.

This paper presents a novel series-parallel reconfiguration circuit with voltage equalization capability. The number of cells connected in series can be changed one by one without causing cell voltage imbalance. Furthermore, the proposed circuit is applicable to any number of cells, realizing flexible design and good modularity. The proposed reconfiguration circuit for the $n$-cell module can operate as a 1S ... $n$-S configuration, further improving the energy utilization ratio compared with [18]. The remainder of this paper is organized as follows: Section 2 introduces the proposed series-parallel reconfiguration circuit; the operation analysis will be performed in Section 3; followed by experimental verification tests for the five-cell module in Section 4.

## 2. Proposed Series-Parallel Reconfiguration Circuit

### 2.1. Circuit Topology

The proposed series-parallel reconfiguration circuit for the five-cell module is shown in Figure 7 as an example. The total switch count is $3(n-1)$, where $n$ is the cell count. The topology is very similar to the conventional balance-shift circuit in Figure 4a, but the conventional circuit cannot be used for modules consisting of the odd number of cells, as explained in Section 1. On the other hand, the proposed reconfiguration circuit is applicable to any number of cells thanks to the novel switching operation, achieving good modularity.

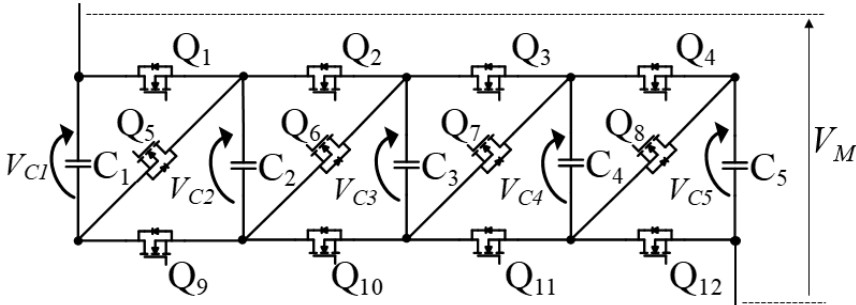

**Figure 7.** Proposed series-parallel reconfiguration circuit for five cells.

### 2.2. Major Features

The proposed reconfiguration circuit is applicable to any number of cells, and, therefore, the design flexibility and extendibility can be improved compared with the conventional balance-shift circuits that can only be applicable to an even number of cells. The generalized form of the proposed reconfiguration circuit is illustrated in Figure 8. For example, the proposed reconfiguration circuit allows changing the cell count from four to five, which is equivalent to a 1.25 times enhancement in module capacitance, achieving fine-tuning in module capacitance. Meanwhile, the cell count in conventional balance-shift circuits (Figures 3 and 4) should be six from four, which is a 1.5 times increase in module capacitance, and their module capacitance cannot be tuned finely.

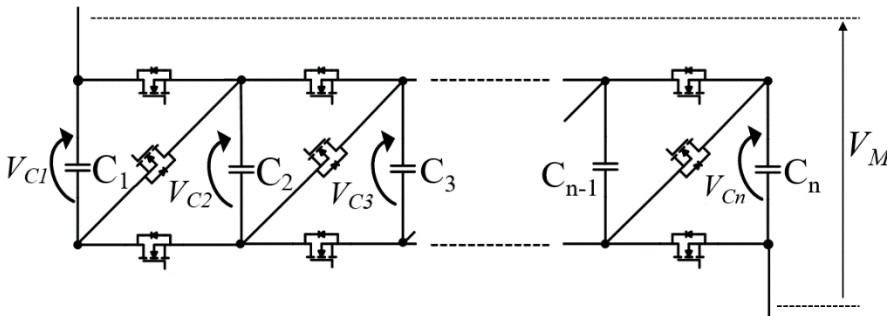

**Figure 8.** Proposed series-parallel reconfiguration circuit for $n$ cells.

All cells in the proposed reconfiguration circuit can charge and discharge uniformly without generating voltage imbalance. Hence, the issue of the conventional unbalance-shift circuit can be resolved, improving the module's energy utilization ratio in comparison with conventional unbalance-shift circuits.

## 3. Operation Analysis

### 3.1. Operation Principle

Cell and module voltage profiles of the proposed reconfiguration circuit are illustrated in Figure 9. The series-parallel connection of cells in the discharging process (Figure 9a) is sequentially reconfigured from 1-series 5-parallel (1S-5P) to 5S-1P configurations, so that

the variation range of the module voltage $V_M$ is between the upper and lower voltage limits of $V_U$ and $V_L$. There are five configurations (i.e., 1S-5P, 2S, 3S, 4S, and 5S-1P), and 2S-, 3S-, and 4S-configurations have their own unique substates. The substates in 2S-, 3S-, and 4S-configurations are repetitively switched to achieve cell voltage equalization. Current flow paths in each configuration or substate in discharging process are shown in Figure 10.

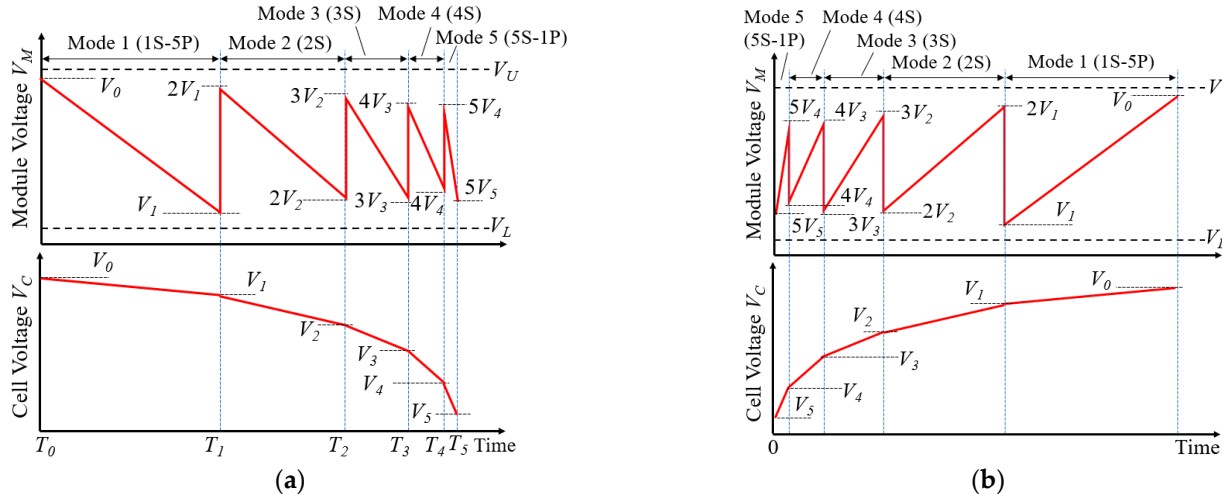

**Figure 9.** Module and cell voltage profiles in (**a**) discharging and (**b**) charging processes.

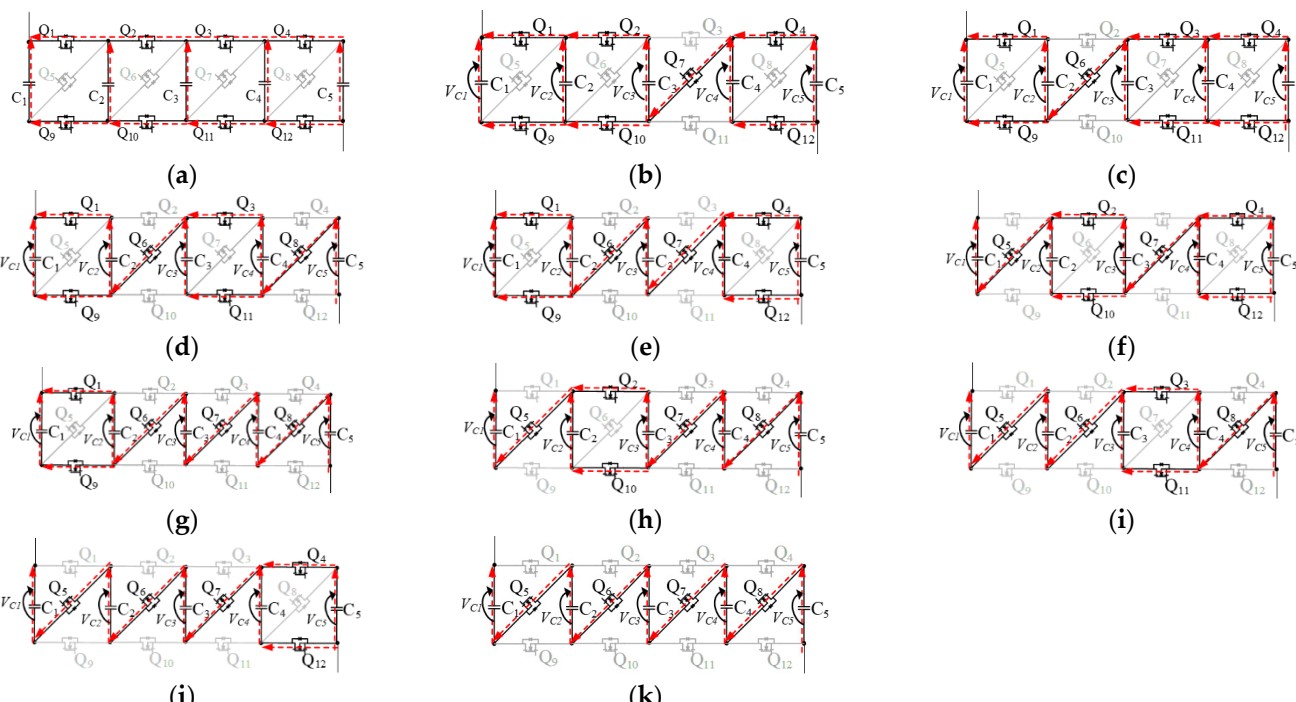

**Figure 10.** Operation modes. (**a**) Mode 1 (1S-5P), (**b**) substate 2A, (**c**) substate 2B, (**d**) substate 3A, (**e**) substate 3B, (**f**) substate 3C, (**g**) substate 4A (**h**) substate 4B, (**i**) substate 4C, (**j**) substate 4D, (**k**) mode 5 (5S-1P).

Mode 1 (1S-5P) (Figure 10a): The module starts discharging with 1S-5P configuration. All cells are connected in parallel through switches and, therefore, cell voltages are uniform. Before $V_M$ reaches $V_L$, the operation mode moves to the 2S configuration.

Mode 2 (2S) (Figure 10b,c): The module configuration alternates between two substates. In substate 2A (Figure 10b), $C_1$–$C_3$ are connected in parallel, while the remaining cells of

$C_4$ and $C_5$ are also connected in parallel in a different group. Their voltage relationships in substate 2A are expressed as:

$$\begin{cases} V_{C1} = V_{C2} = V_{C3} \\ \qquad V_{C4} = V_{C5} \end{cases} \text{(Substate 2A)} \tag{2}$$

where $V_{C1}$–$V_{C5}$ are the voltages of $C_1$–$C_5$. The combined capacitance of $C_1$–$C_3$ is 1.5 times larger than that of $C_4$–$C_5$, generating voltage imbalance. In substate 2B (Figure 10c), on the other hand, $C_3$ is connected in parallel with $C_4$ and $C_5$, yielding the following voltage relationships:

$$\begin{cases} \qquad V_{C1} = V_{C2} \\ V_{C3} = V_{C4} = V_{C5} \end{cases} \text{(Substate 2B)} \tag{3}$$

Since the combined capacitances of $C_1$–$C_2$ and $C_3$–$C_5$ are different, their voltages tend to be mismatched. Although both substates contain unbalanced combined capacitances, all cells are virtually connected in parallel by switching substates 2A and 2B repetitively at a fixed frequency. Repetitive switching between substates 2A and 2B is equivalent to combining (2) and (3), yielding $V_{C1} = V_{C2} = V_{C3} = V_{C4} = V_{C5}$. Thus, the virtual parallel connection equalizes all cell voltages in the module.

Mode 3 (3S) (Figure 10d–f): The series-parallel configuration alternates among three substates. Either $C_1$, $C_3$, or $C_5$ is discharged alone, whereas the remaining cells are discharged in parallel with another cell. In substate 3A (Figure 10d), for example, the discharging current of $C_5$ is twice that of other cells. Similarly, a larger current flows through $C_3$ and $C_1$ in substates 3B and 3C, respectively. Although cell voltages tend to be imbalanced in these substates in 3S configuration, repetitively alternating the configuration among substates 3A–3C realizes the virtual parallel connection for all cells. From Figure 10d–f, the voltage relationships in substates 3A–3C are:

$$\begin{cases} V_{C1} = V_{C2} \\ V_{C3} = V_{C4} \end{cases} \text{(Substate 3A)} \tag{4}$$

$$\begin{cases} V_{C1} = V_{C2} \\ V_{C4} = V_{C5} \end{cases} \text{(Substate 3B)} \tag{5}$$

$$\begin{cases} V_{C2} = V_{C3} \\ V_{C4} = V_{C5} \end{cases} \text{(Substate 3C)} \tag{6}$$

Combining (4)–(6) produces $V_{C1} = V_{C2} = V_{C3} = V_{C4} = V_{C5}$, achieving the voltage equalization.

Mode 4 (4S) (Figure 10g–j): There are four substates in 4S configuration. Two out of five cells are connected in parallel, while other cells are connected in series. In substate 4A (Figure 10g), for example, $C_1$ and $C_2$ are connected in parallel, and other cells are connected in series. In other substates, the other two cells are connected in parallel. Cell voltage relationships in substates 4A–4D (Figure 10g–j) are expressed as:

$$V_{C1} = V_{C2} \;\text{(Substate 4A)} \tag{7}$$

$$V_{C2} = V_{C3} \;\text{(Substate 4B)} \tag{8}$$

$$V_{C3} = V_{C4} \;\text{(Substate 4C)} \tag{9}$$

$$V_{C4} = V_{C5} \;\text{(Substate 4D)} \tag{10}$$

From (7)–(10), $V_{C1} = V_{C2} = V_{C3} = V_{C4} = V_{C5}$. Similar to the 2S and 3S configurations, repetitive alternation among these four substates realizes cell voltage equalization.

Mode 5 (5S-1P) (Figure 10k): All cells are connected in series and discharge uniformly without generating voltage imbalance.

The mode transition is illustrated in Figure 11. The discharging process starts with Mode 1 (1S-5P configuration) and ends with Mode 5 (5S-1P configuration). Substates

in each configuration are repetitively switched to prevent the occurrence of cell voltage imbalance. In the charging process, series-parallel configurations are shifted in the opposite order (i.e., from Mode 5 to Mode 1) so that the variation range of $V_M$ is between $V_U$ and $V_L$, as illustrated in Figure 9b.

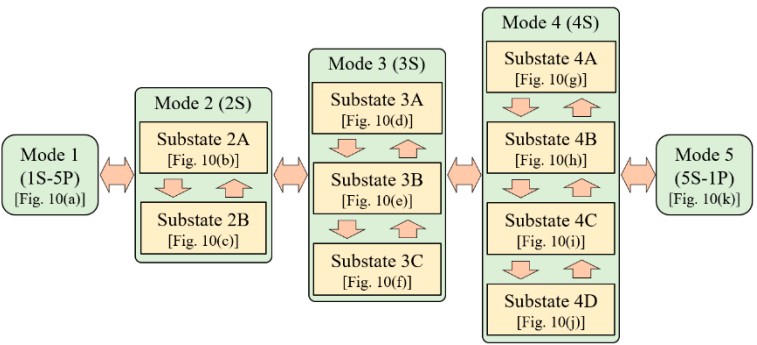

**Figure 11.** Mode transition map.

### 3.2. Operation Condition

The series-parallel connection of cells is reconfigured depending on cell voltages. The module voltage $V_M$ must stay between the upper limit $V_U$ and lower limit $V_L$, even when the circuit is reconfigured. The cell and module voltage profiles in Figure 9a yield the following equations:

$$V_U \geq \begin{cases} V_0 & (t = T_0) \\ 2V_1 & (t = T_1) \\ 3V_2 & (t = T_2) \\ 4V_3 & (t = T_3) \\ 5V_4 & (t = T_4) \end{cases} \tag{11}$$

$$V_L \leq \begin{cases} V_1 & (t = T_1) \\ 2V_2 & (t = T_2) \\ 3V_3 & (t = T_3) \\ 4V_4 & (t = T_4) \\ 5V_5 & (t = T_5) \end{cases} \tag{12}$$

where $T_0$–$T_5$ are the time at the beginning or end of operation modes in the discharging process (see Figure 9a). Rearrangement of (11) and (12) produces:

$$\frac{V_U}{2} \geq V_1 \geq V_L, \qquad \frac{V_U}{3} \geq V_2 \geq \frac{V_L}{2}, \qquad \frac{V_U}{4} \geq V_3 \geq \frac{V_L}{3}, \qquad \frac{V_U}{5} \geq V_4 \geq \frac{V_L}{4} \tag{13}$$

These equations can also be applied to the charging process (Figure 9b).

### 3.3. Reconfiguration Algorithm

Assuming all cell voltages are equalized to be $V_C$, the operation modes are determined, based on $V_C$, as:

$$\begin{cases} V_C \geq V_1 & \rightarrow (1\text{S} - 5\text{P in Mode 1}) \\ V_1 > V_C \geq V_2 & \rightarrow (2\text{S in Mode 2}) \\ V_2 > V_C \geq V_3 & \rightarrow (3\text{S in Mode 3}) \\ V_3 > V_C \geq V_4 & \rightarrow (4\text{S in Mode 4}) \\ V_4 > V_C \geq V_5 & \rightarrow (5\text{S} - 1\text{P in Mode 5}) \end{cases} \tag{14}$$

The flowchart of the reconfiguration sequence for the five-cell module is shown in Figure 12. Series-parallel configuration is simply determined based on the measured $V_C$. Substates in Modes 2–4 are switched at several hertz to preclude the occurrence of cell voltage imbalance.

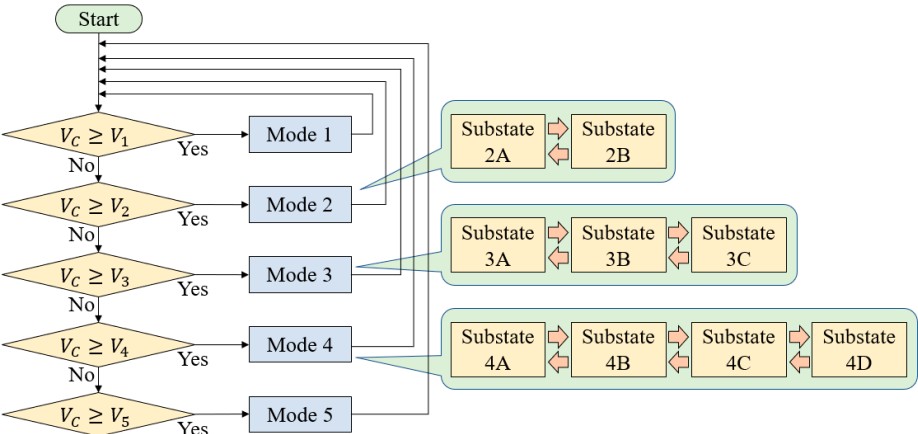

**Figure 12.** Flowchart of reconfiguration sequence.

*3.4. Switching Frequency*

Although substates in Modes 2–4 repetitively switch, cell voltage imbalance occurs to some extent in each substate due to unbalanced combined capacitances, as discussed in Section 3.1. A huge cell voltage imbalance might trigger an excessively large current at switching moments. A switching frequency $f$ should be properly determined to prevent the occurrence of a large cell voltage imbalance. In the following, the voltage imbalance between $C_4$ and $C_5$ in Mode 3 is focused as an example.

In substate 3A (Figure 10d), $C_5$ discharges alone, while $C_4$ shares the discharge current with $C_3$. Assuming all cell capacitances are $C$, the current difference between $C_4$ and $C_5$ is $I_M/2$, where $I_M$ is the module's discharge current. The voltage imbalance generated in substate 3A, $\Delta V$, is expressed as:

$$\Delta V = \frac{I_M}{2Cf} \tag{15}$$

where $f$ is the switching frequency. In the next substate (substate 3B in Figure 10e), $C_4$ and $C_5$ are connected in parallel, and a balance current $I_{eq}$ flows between them. Cells' internal resistances $r_4$ and $r_5$, and switches' on-resistance $R_{on}$ limit $I_{eq}$ are expressed as:

$$I_{eq} = \frac{\Delta V}{r_4 + r_5 + 2R_{on}} = \frac{\Delta V}{R_{total}} \tag{16}$$

where $R_{total}$ is the total resistance. In general, $R_{total}$ is in the range of several to a few ten milliohms and, therefore, $\Delta V$ should be within a few ten millivolts. Equation (15) suggests that $f < 10$ Hz would be high enough to achieve $\Delta V < 10$ mV for 400 F cells, even when $I_M$ ranges several ten amperes.

Although Mode 3 was focused above, operations in Modes 2 and 4 can be analyzed similarly. Since the numbers of substates in each mode are different (i.e., two, three, and four substates in Modes 2, 3, and 4, respectively), $f$ is individually set for each mode. In Mode 3, for example, it takes four steps before coming back to the original substates (i.e., 3A → 3B → 3C → 3B → 3A). Modes 2 and 4, on the other hand, require two steps (2A → 2B → 2A) and six steps (4A → 4B → 4C → 4D → 4C → 4B → 4A), respectively. The switching frequency $f$ was determined to be 2.5, 5, and 10 Hz for Modes 2, 3, and 4, respectively, so that whole steps in each operation mode are completed in a similar time range; whole steps in Mode 2, 3, and 4 are completed in 0.8 s (=2 steps/2.5 Hz), 0.8 s (=4 steps/5 Hz), and 0.6 s (=6 steps/10 Hz), respectively.

## 4. Experimental Results

The proposed reconfiguration circuit for five cells was built, as shown in Figure 13. 400 F EDLC cells with a rated charge voltage of 2.5 V were used. TMS320F28335 control card (Texas Instruments, Dallas, TX, USA) was used to measure cell voltages, implement

the flowchart of the reconfiguration algorithm (Figure 12), and generate driving signals for switches. N-channel MOSFETs (IRFR3410, $R_{on}$ = 39 mΩ) were used as switches. The gate-source voltage of 10 V was applied to turn on switches. Substates in 2S-, 3S-, and 4S-configurations were switched at 2.5, 5, and 10 Hz, respectively. $V_1$–$V_4$ were set to be 1.25, 0.83, 0.625, and 0.5 V, respectively. The EDLC module was cycled with a constant current of 2.0 A.

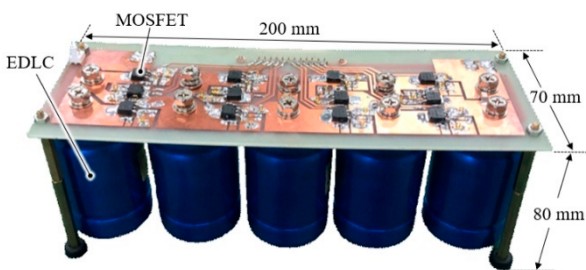

**Figure 13.** Prototype of series-parallel reconfiguration circuit for five EDLC cells.

The measured discharging and charging voltage profiles are shown in Figure 14a,b, respectively. The module started discharging in Mode 1 (1S-5P configuration) and was switched to Mode 2 (2S configuration) when $V_C$ reached $V_1$ = 1.25 V. Cell voltage profiles in Mode 2 are magnified in the inset of Figure 14a. The voltage of $C_5$ was slightly higher than others. This offset is attributable to a mismatch in cells' internal resistances, and the internal resistance of $C_5$ would have been lower than others—the voltage of $C_5$ in Mode 2 was the highest and lowest in discharging and charging processes in Figure 14a,b, respectively. However, the voltage imbalance was as low as 10 mV, and all cell voltages were satisfactorily balanced. All cells continued to uniformly discharge in Modes 3 and 4 (3S- and 4S-configurations) without causing cell voltage imbalance. All cells were discharged as low as 0.3 V at the end of the discharging experiment, which was equivalent to the energy utilization ratio of 98.6%.

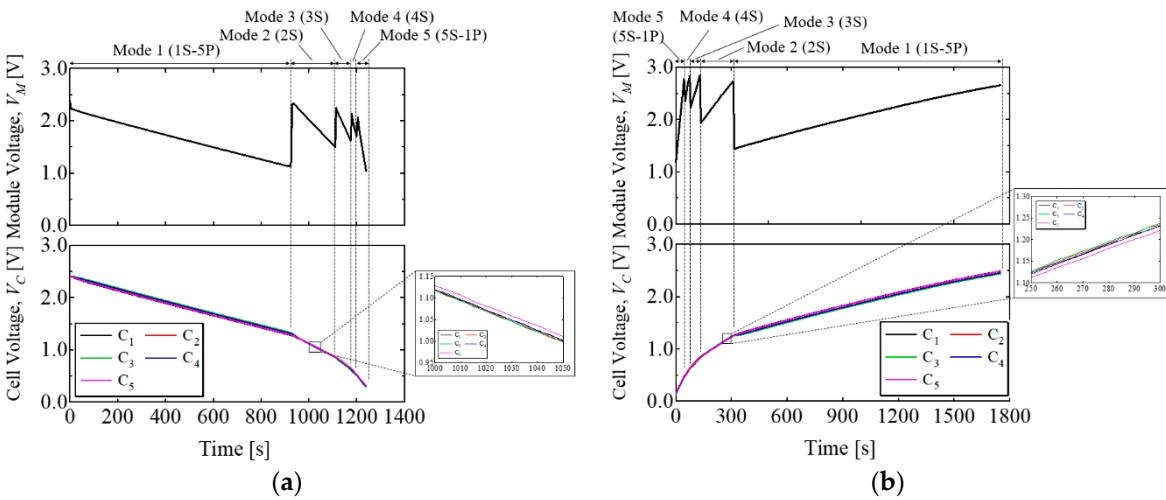

**Figure 14.** Resultant module and cell voltage profiles in (**a**) discharging and (**b**) charging processes.

The variation range of $V_M$ was limited within 1.04 V and 2.83 V, while all cells discharged from 2.5 V to 0.3 V. Both discharging and charging voltage profiles matched very well with the theoretical characteristics, demonstrating the proposed reconfiguration technique.

## 5. Conclusions

This paper has proposed the series-parallel reconfigurable EDLC module with cell voltage equalization capability. The series-parallel connection of EDLC cells is changed

so that the module voltage variation can be within the desired range. The number of series-connected cells can be changed one by one, achieving finer voltage step changes than conventional reconfiguration techniques. The proposed reconfiguration circuit is applicable to any number of cells, and the design flexibility and extendibility can be improved in comparison with conventional techniques. Furthermore, the proposed reconfiguration circuit can equalize cell voltages, and all cells can be cycled uniformly, achieving improved energy utilization of the EDLC module.

A prototype of the five-cell module was built and tested. The resultant cell and module voltage profiles matched well with the theoretical characteristics. All cells were uniformly discharged from 2.5 V to 0.3 V, while the module voltage varied between 1.04 and 2.83 V. This result is equivalent to a 98.6% energy utilization ratio.

**Author Contributions:** Conceptualization, M.U. and Z.L.; methodology, M.U. and Z.L.; software, Z.L.; validation, Z.L. and K.K.; formal analysis, M.U.; writing—original draft preparation, M.U.; writing—review and editing, M.U.; supervision, M.U. and K.K.; project administration, M.U. All authors have read and agreed to the published version of the manuscript.

**Funding:** This research received no external funding.

**Institutional Review Board Statement:** Not applicable.

**Informed Consent Statement:** Not applicable.

**Conflicts of Interest:** The authors declare no conflict of interest.

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
