# Peer review of "Series-Parallel Reconfigurable Electric Double-Layer Capacitor Module with Cell Equalization Capability, High Energy Utilization Ratio, and Good Modularity"

_energies, doi:10.3390/en14123689_

Round 1

Reviewer 1 Report

This paper proposed a series-parallel reconfigurable EDLC module to achieve a balance-shift circuit when the cell number was odd (e.g., 5). The author pointed out the importance of this work and developed a series of modules with a balance-shift circuit. This work is well-organized and investigated. I recommend publication without further modification.

Author Response

Dear Reviewer,

We are very grateful to you for your thoughtful and helpful review of the manuscript. Your comments and suggestions have been incorporated as appropriate into the revised manuscript. The revised parts are highlighted with green in the revised manuscript. 

Reviewer 2 Report

In this manuscript, the authors reported series-parallel reconfigurable EDLC module with cell voltage equalization capability. The cell number did not have any limitation, realizing flexible module design. The study is decent, whereas some problems exist. Therefore, I recommend major revisions before the acceptance of this manuscript in Energies. My detailed comments are as follows:

  1. The Abstract and Introduction section, including language and figures, are highly similar with the paper published by the same group in 2019 (Energies 2019, 12, 2741). Despite similar research topic, this phenomenon should be avoided.
  2. The meaning of VC1, VC2, VC3, and VC4 in the Figure 5 need to be indicated.
  3. How many times were the cell voltage profiles tested (Figure 14)?
  4. What will the cell voltage profiles be like if increase the discharge and charge current?

Author Response

(The authors gave the same response as above.)

Reviewer 3 Report

The authors presents a series-parallel reconfigurable EDLC module with cell voltage equalization capability. This EDLC module can be applied to any number of cells, realizing good modularity, its idea was proved by a prototype. Overall, this paper is well-written and the motivation of the proposed scheme is clear. The reviewer has some following comments to improve the quality of this paper.

1. One of motivations in your scheme is to apply any number of cells to improve the disadvantage of balance-shift circuit. Why do we need to use the odd number of cells with the balanced cell voltage? If the one of reasons is to increase module capacity, we would just use even number of cells with high capacity?

2. Please express C1~C4 in figure 3(a) and 5.(a) for the readers' understanding.

3. In your proposed scheme, the sub-states in 2S-, 3S-, and 4S-configurations are repeatedly switched to achieve cell voltage equalization. Please describe how to achieve cell voltage equalization by switch repetition in detail.
If you explain by using the operating and voltage diagram in these sub-states, this would be helpful to readers.

4. What controls the switch of this module including the sub-state and how did the authors implement the overall controller? The authors described the flowchart in figure 12, but there is no explanation for the its controller implementation.

5. How do you determine the value of the switching frequencies used in the sub-states of modes 2-4? In this prototype, you used 2.5, 5, and 10Hz in 2S-, 3S-, and 4S-configurations. Is there any reason to determine these frequencies and what if the faster or slower frequencies are used?

6. In the prototype, the authors used the N-channel MOSFET as a switch. Why do you use the N-channel MOSFET? You could have used both the N-channel and P-channel MOSFETs, or the P-channel MOSFET. And what is the value of the switch voltage? (the gate voltage of the N-channel MOSFET)   

7. In figure 14, it seems there is a little offset in C5. what is the reason for this offset, and is there any negative impact on the module performance?

8. In reference section, some numbers have error, please check and modify it.

Author Response

(The authors gave the same response as above.)

Round 2

Reviewer 2 Report

The authors have revised all the problems in the manuscript. Therefore, I recommend the acceptance of this manuscript in Energies.

Reviewer 3 Report

The authors have addressed all of the reviewer's questions and demands. There are no more comments and this paper got better than the first version.